# AREEBA: An Area Efficient Binary Huff-Curve Architecture

Asher Sajid [1], Muhammad Rashid [2] , Sajjad Shaukat Jamal [3] , Malik Imran [4,5,*] and Saud S. Alotaibi [6] and Mohammed H. Sinky [2]

1    Department of Electrical Engineering, Bahria University, Islamabad 44000, Pakistan; malikasher267@gmail.com
2    Department of Computer Engineering, Umm Al-Qura University, Makkah 24382, Saudi Arabia; mfelahi@uqu.edu.sa (M.R.); mhsinky@uqu.edu.sa (M.H.S.)
3    Department of Mathematics, College of Science, King Khalid University, Abha 61413, Saudi Arabia; shussain@kku.edu.sa
4    Science and Technology Unit (STU), Umm Al-Qura University, Makkah 24351, Saudi Arabia
5    Department of Computer Systems, Tallinn University of Technology, 12616 Tallinn, Estonia
6    Department of Information Systems, Umm Al-Qura University, Makkah 24351, Saudi Arabia; ssotaibi@uqu.edu.sa
*    Correspondence: malik.imran@taltech.ee; Tel.:+372-53676608

**Abstract:** Elliptic curve cryptography is the most widely employed class of asymmetric cryptography algorithm. However, it is exposed to simple power analysis attacks due to the lack of unifiedness over point doubling and addition operations. The unified crypto systems such as Binary Edward, Hessian and Huff curves provide resistance against power analysis attacks. Furthermore, Huff curves are more secure than Edward and Hessian curves but require more computational resources. Therefore, this article has provided a low area hardware architecture for point multiplication computation of Binary Huff curves over $GF(2^{163})$ and $GF(2^{233})$. To achieve this, a segmented least significant digit multiplier for polynomial multiplications is proposed. In order to provide a realistic and reasonable comparison with state of the art solutions, the proposed architecture is modeled in Verilog and synthesized for different field programmable gate arrays. For Virtex-4, Virtex-5, Virtex-6, and Virtex-7 devices, the utilized hardware resources in terms of hardware slices over $GF(2^{163})$ are 5302, 2412, 2982 and 3508, respectively. The corresponding achieved values over $GF(2^{233})$ are 11,557, 10,065, 4370 and 4261, respectively. The reported low area values provide the acceptability of this work in area-constrained applications.

**Keywords:** elliptic curve cryptography; Binary Huff curves; area optimization; crypto processor; field programmable gate array (FPGA)

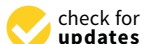



## 1. Introduction

Elliptic-curve cryptography (ECC), a public-key cryptography scheme, has become an attractive approach to provide security for area-constrained applications like internet-of-things (IoT), radio-frequency-identification (RFID) cards, digital signatures and many more [1]. Its widely adoption is due to its ability to provide a similar security level with relatively smaller key-sizes as compared to the current public-key security standard (i.e., the Rivest–Shammir–Adleman (RSA) algorithm) [2]. From structural point of view, ECC involves four layers of operations [3]. The top most layer (fourth layer) ensures the encryption and decryption of data. In layer three, the point multiplication (PM) is computed which is the most critical operation. For the PM computation, the required point addition (PA) and point doubling (PD) operations are employed in layer two. Finally, the layer one of ECC consists of finite-filed (FF) arithmetic operations (addition, multiplication, square and inversion). In addition to the layer model of ECC, there are two coordinate systems, i.e., affine, and projective. In past, the latter has more frequently been employed to optimize throughput [3]. Furthermore, two field representations, i.e., prime ($GF(\mathbb{P})$) and binary

($GF(2^m)$), are commonly involved. The prime field representation is generally utilized for software-based implementations while the binary field representation is preferred for hardware deployments [3,4].

Concerning the security strength of ECC, several implementation models such as Weierstrass [3,4], Binary-edward (BEC) [5], Hessian (HC) [6], and Binary-huff (BHC) [7] are frequently utilized. The fundamental model (Weierstrass) in ECC allows access to attackers and, therefore, exposes the secret-key via simple power attacks (SPA) [5–7]. The vulnerability of Weierstrass model to SPA is due to different mathematical formulations of PA and PD for PM computation [8]. The SPA is a type of side channel attack where attackers can find the secret-key in terms of zeros and ones by inspecting the power trails of PA and PD computations. The power trails are inspected using various power analysis tools such as logic analyzer. Among several other choices, one of the solutions to resist SPA in ECC is the use of unified PA and PD laws (*Unif_Add_Law*).

In addition to the Weierstrass model, all the aforesaid models (HC, BEC, BHC) provide Unif_Add_Law for the computation of PA and PD operations [5–7]. Furthermore, the computational complexity of PM operation in BEC and HC models are lower than BHC [5,6]. Therefore, the BEC and HC models of ECC are generally preferred for those applications where a higher throughput is more important than any other design parameter. On the other hand, the BHC model provides higher security as compared to BEC and HC models [7]. The BHC is a binary extension of Huff's model with a unified point addition and doubling [9]. For complete mathematical structures of various *Unif_Add_Law* models, the interested readers are referred to [5–7,9]. Moreover, we also refer to [10–13] where more generalizations and theoretical foundations over Weierstrass, Hessian, Huff and Edward models have been presented. At the same time, the major applications for highly secure public key cryptosystems against SCA with low area demands include cloud computing [14], Identity-Based Encryption [15], wireless sensor networks (WSNs) [16] and RFID [17] etc. Therefore, the purpose of this article is to provide a low area implementation of BHC model for ECC on a reconfigurable field programmable gate array (FPGA).

### 1.1. Existing FPGA Architectures and Limitations

To provide resistance against SPA using the BHC model, only a limited number of hardware architectures are available. The reason is that the BHC model was recently proposed in 2011 [18–22]. For other unified ECC models (such as BEC), there are several architectures. However, due to different mathematical structures for PA and PD computations in BEC and BHC, a fair comparison is not realistic. Therefore, it is important to mention that this article mainly considers BHC architectures for discussion. Nevertheless, for the sake of completeness, some interesting hardware accelerators for the BEC model are also discussed [8,23–25].

The first hardware architecture for the PM computation of the BHC model over $GF(2^{233})$ is described in [18]. The critical FF arithmetic operations (i.e., multiplication and inversion) are computed by employing a hybrid Karatsuba multiplier and an Itoh–Tsujii inversion algorithm, respectively. Furthermore, to achieve a hybrid multiplication architecture, the simple and general Karatsuba multipliers are coupled. The general Karatsuba multiplier is employed for better utilization of FPGA look-up-tables (LUTs) over smaller bits. On the other hand, the simple Karatsuba multiplier is used to minimize gate counts over longer bits. In order to reduce the required number of clock cycles, a Quad block Itoh–Tsujii algorithm is implemented. As a result, the entire PM architecture utilizes 20437 FPGA slices on Virtex-4 FPGA. Similarly, another FPGA-based architecture over $GF(2^{233})$ is provided in [19]. The main contribution of [19] is to report SPA attacks and countermeasures for PA and PD operations in BHC model of ECC. However, on the newer Virtex-7 FPGA by Xillinx, the architecture utilizes 6032 hardware resources (in terms of slices).

Recently, the pipelined architectures for BHC model of ECC have been presented in [20–22]. An efficient throughput/area architecture is presented in [20]. The pipelining is used to reduce the critical path and to improve the clock frequency of the PM architecture.

To avoid structural hazards, which occur due to pipelining, the scheduling for PA and PD instructions are given. Moreover, for modular multiplication, a parallel least significant digit multiplier is employed. The implementation results over $GF(2^{163})$ and $GF(2^{233})$ are provided on a Virtex-7 FPGA device. The architecture utilizes 6342 FPGA slices and achieves a maximum clock frequency of 369 MHz over $GF(2^{233})$. A four-stage pipelined architecture is given in [21]. It reduces the hardware cost of the BHC model by revisiting the original PA and PD mathematical formulations. As a result, the simplified mathematical formulations is presented with a 43% reduction in hardware resources. Towards the throughput optimization, pipeline registers are incorporated in datapath. Moreover, in order to reduce clock cycles, an efficient scheduling for the computation of PA and PD instructions are provided. Considering the field length of $GF(2^{233})$, the synthesis results are given for Virtex-7 FPGA.

The architectures in [18–21] are specifically proposed for the Unif_Add_Law computations of PM operation. It implies that all the optimizations are made for a single ECC model (which is BHC). On the other hand, the architecture in [22] provides a flexible solution and supports two different models of ECC (BHC and Weirstrass). Additionally, the architecture utilizes two distinct PM algorithms for different ECC models. The Double and Add algorithm is used for the BHC model of ECC while the Montgomery ladder algorithm is utilized for Weirstrass model. It provides three different designs, i.e., a dedicated architecture for each BHC model and Weirstrass model and a flexible design to integrate BHC with Weirstrass. The dedicated architecture of [22] for BHC model utilizes 6083 FPGA slices and require 36 μs for one PM computation. The BEC architecture of [23] utilizes 21816 FPGA slices and achieves a clock frequency of 48 MHz. Another BEC architecture, reported in [24], consumes 5919 FPGA slices on Virtex-5 FPGA. Similarly, the BEC architectures reported in [8,25] utilize 15804 and 6600 FPGA slices, respectively.

It can be observed from the above discussion that the required hardware resources (FPGA slices) in existing FPGA implementations of BHC and BEC models are relatively higher [8,18–25]. The architectures resulting higher hardware resources are not suitable for area constrained applications such as Identity-Based Encryption [15], WSNs [16] and RFID [17]. Consequently, a low area implementation of BHC model is required for area constrained applications.

### 1.2. Settings and Contributions

Our settings (basic representation and coordinate system) as well as contributions for the computation of PM operation in the BHC model of ECC are given as:

- **Settings:**
    - **Basis representation:** This work employs a polynomial basis representation instead of a normal basis [18–22]. It is important to note that the BHC model of ECC requires frequent multiplications. To achieve frequent multiplications, the polynomial basis representation is more useful while the normal basis is convenient where repeated squarings are needed.
    - **Coordinate system:** An affine coordinate system requires an FF inversion during each PA and PD computation [7,18–22] which ultimately effects the latency of entire architecture. Consequently, to avoid the cost of FF inversion required during each computation of PA and PD operation, a projective (Lopez Dahab) coordinate system is selected in this article.

- **Contributions:**

    - **PM architecture:** We have presented a PM architecture with reduced area over $GF(2^m)$ for $m = 163$ and 233 bits (details are given in Section 3).
    - **Polynomial multiplier architecture:** Towards area reduction, we have proposed a segmented least significant digit (segmented-LSD) multiplier with a digit size of $d = 32$-bits. Each created digit is segmented into four 8-bit digits. Subsequently, the multiplication over each 8-bit segment is performed by using a

> simple schoolbook multiplication method. This ultimately reduces the hardware resources (in terms of FPGA slices).
> – **Dedicated controller:** To efficiently control the inserted logic for the proposed low area PM architecture, a finite state machine (FSM)-based dedicated controller is employed.

Our contributions in this article have resulted in an area optimized architecture for PM computation of BHC model over $GF(2^{163})$ and $GF(2^{233})$. The proposed architecture is termed AREEBA (*Are*a *E*fficient *B*inary Huff-curve *A*rchitecture). The italic characters present our selection for the acronym AREEBA. The architecture is modeled in Verilog (HDL) using the Xilinx ISE design tool. To provide a realistic and reasonable comparison with state of the art solutions, published in [8,18–25], the proposed architecture is synthesized for various FPGA devices. For Virtex-4, Virtex-5, Virtex-6 and Virtex-7 devices, the utilized FPGA slices over $GF(2^{163})$ are 5302, 2412, 2982 and 3508, respectively. The corresponding required hardware resources (in terms of FPGA slices) over $GF(2^{233})$ are 11,557, 10,065, 4370 and 4261, respectively. On newer Virtex-7 FPGA, the proposed architecture provides 1.41, 1.48, 1.64 and 1.42 times lower hardware resources as compared to BHC architectures, published in [19–22], respectively. Moreover, the proposed architecture is 31.79 times faster (in terms of computation time) as compared to the most recent Twisted BEC architecture, reported in [8] (implemented on Virtex-6 and constructed over $GF(\mathbb{P})$). It is important to note that the proposed architecture requires relatively higher clock cycles (10,393 for $GF(2^{163})$ and 11,137 for $GF(2^{233})$) with a lower clock frequency. Nevertheless, the optimized area values reveal that the presented architecture is suitable for area-constrained environments.

The structure of this work is given as follows: The mathematical background for the implementation of BHC model over $GF(2^m)$ is presented in Section 2. Subsequently, Section 3 describes the proposed hardware architecture. The implementation results and comparison with state of the art is provided in Section 4. Finally, Section 5 concludes the paper.

## 2. Preliminaries

This section describes the required mathematical background pertaining to BHC model of ECC over $GF(2^m)$, along with the corresponding formulations required for the computation of $Unif\_Add\_Law$ in BHC over $GF(2^m)$ (Section 2.1). Furthermore, the details for the implemented Double and Add PM algorithm are presented in Section 2.2.

### 2.1. BHC Over $GF(2^m)$

Binary Huff curve is defined as the set of projective points $(X : Y : Z)$ over $GF(2^m)$ which satisfy the following equation:

$$E : aX(Y^2 + fYZ + Z^2) = bY(X^2 + fXZ + Z^2) \tag{1}$$

In Equation (1), the variables $a$, $b$ and $f$ are curve parameters which belong to $GF(2^m)$ while considering $a \neq b$.

Initially, the Huff model was introduced in 1963 [26]. In 2010, the descriptions and formulations of odd characteristic fields with an outline for the binary field were reported [9]. Thereafter, in 2011, the formal construction of the Huff model for binary field was provided [7]. This construction provided the first $Unif\_Add\_Law$ for the BHC model of ECC, as given in Table 1.

The $Unif\_Add\_Law$, shown in Table 1, was evaluated in [19]. It was identified that the $Unif\_Add\_Law$, published in [7], shows behavioral differences when computing point addition and double operations. It increases vulnerability to SPA attacks. Based on this observation, another $Unif\_Add\_Law$ is presented in [19]. The corresponding mathematical formulations are presented in Table 2.

**Table 1.** $Unif\_Add\_Law$ for BHC (published in [7]).

| $Unif\_Add\_Law$ of the BHC Model of ECC |
|---|
| $m_1 = X_1 \times X_2, m_2 = Y_1 \times Y_2, m_3 = Z_1 \times Z_2$ |
| $m_4 = (X_1 + Z_1) \times (X_2 + Z_2) + m_1 + m_3$ |
| $m_5 = ((Y_1 + Z_1) \times (Y_2 + Z_2)) + m_2 + m_3, m_6 = m_1 \times m_3, m_7 = m_2 \times m_3$ |
| $m_8 = m_1 \times m_2 + m_3^2, m_9 = m_6(m_2 + m_3)^2, m_{10} = m_7(m_1 + m_3)^2$ |
| $m_{11} = m_8 \times (m_2 + m_3), m_{12} = m_8 \times (m_1 + m_3)$ |
| $X_3 = \alpha \times m_9 + m_4 \times m_{11}$ |
| $Y_3 = \beta \times m_{10} + m_5 \times m_{12}$ |
| $Z_3 = m_{11} \times (m_1 + m_3)$ |

**Table 2.** $Unif\_Add\_Law$ for BHC (published in [19]).

| $Unif\_Add\_Law$ of the BHC Model of ECC |
|---|
| $m_1 = X_1 \times X_2, m_2 = Y_1 \times Y_2, m_3 = Z_1 \times Z_2$ |
| $m_4 = (X_1 + Z_1) \times (X_2 + Z_2), m_5 = ((Y_1 + Z_1) \times (Y_2 + Z_2))$ |
| $m_6 = m_1 \times m_3, m_7 = m_2 \times m_3, m_8 = m_1 \times (m_2 + m_3)$ |
| $m_9 = m_6(m_2 + m_3)^2, m_{10} = m_7(m_1 + m_3)^2, m_{11} = m_8 \times (m_2 + m_3)$ |
| $X_3 = \alpha \times m_9 + m_4 \times m_{11} + Z_3$ |
| $Y_3 = \beta \times m_{10} + m_5 m_8 \times (m_1 + m_3) + Z_3$ |
| $Z_3 = m_{11} \times (m_1 + m_3)$ |

The terms $X_3$, $Y_3$, and $Z_3$, used in Tables 1 and 2, are the projective coordinates of the points on the defined Huff curve. The additional terms $\alpha$ and $\beta$ are constants defined as, $\alpha = \frac{(a+b)}{b}$ and $\beta = \frac{(a+b)}{a}$. There are two different possibilities to use these constant parameters: precomputed and runtime computation. In this work, the precomputed curve constants, i.e., $\alpha$ and $\beta$ are used to reduce required clock cycles. For hardware implementations, the mathematical formulations of Table 1 are considered in [18]. Similarly, the hardware architectures for mathematical formulations of Table 2 are considered in [19–21]. In this work, the PM computation is implemented using the mathematical formulations from Table 2. It is important to note that the existing architectures of $Unif\_Add\_Law$ over BHC and BEC, published in [8,19,20,23], cover SPA at algorithmic level only. Therefore, this work also considers the SPA at algorithmic level for a fair performance comparison.

*2.2. PM Over $GF(2^m)$*

The PM operation over $GF(2^m)$ is defined as:

$$Q = k \cdot P := \underbrace{P + P + \ldots + P}_{k\text{-times}} \qquad (2)$$

In Equation (2), $Q$ is the resultant point on the BHC curve, $k$ determines the scalar multiplier and $P$ is an initial point on the BHC curve. We employed the following Double and Add PM algorithm (Algorithm 1).

In Algorithm 1, the $Unif\_Add\_Law$ represents a set of equations for PM computation (Tables 1 and 2). It computes $Unif\_Add\_Law$ based on the inspected key bit, i.e., $k$. The $k_{n-1}, \ldots, k_1, k_0$ show the key values in terms of 0 s and 1 s. Similarly, the variable $n$ determines the corresponding key bit position. It is important to note that the length for $n$ and $m$ are same. Therefore, the statements before *for loop* are for initialization (affine to projective conversions). Similarly, the statements inside the *for loop* are for PM computation in projective (Lopez Dahab) coordinates. Finally, the statements after the *for loop* are for projective to affine coordinates conversion.

---

**Algorithm 1:** Double and Add the PM Algorithm (previously employed for BHC in [18–22])

**Input:** $k = (k_{n-1}, \ldots, k_1, k_0)$ with $k_{n-1} = 1$, $P = (x, y) \in GF(2^m)$
**Output:** $Q = k.(P)$

---

$X_1 = x_p$, $Y_1 = Y_2 = y_p$, $Z_1 = 1$, $X_2 = x_p^4 + b$, $Z_2 = x_p^2$
**for** *(i from m-1 down to 0)* **do**
    $Q = Unif\_Add\_Law(Q, Q)$
    **if** $k_i = 1$, **then**
        $Q = Unif\_Add\_Law(P, Q)$
    **end if**
    **end for**
$Return : (x_q, y_q) = (\frac{X_2}{Z_2}, \frac{Y_2}{Z_2^2})$

---

## 3. Proposed Hardware Architecture

The proposed hardware architecture (AREEBA) for BHC model of ECC is illustrated in Figure 1. For testing and verification, the initial BHC curve parameters for the proposed design are selected from National Institute of Standards and Technology (NIST) [27]. The PA and PD formulas from Table 2 are employed to compute PM operation over $GF(2^{163})$ and $GF(2^{233})$. As shown in Figure 1, the proposed architecture consists of a memory unit (MU), Arithmetic Unit (AU) and a dedicated control unit (CU). The additional multiplexers (i.e., M3 and M4) are used for the routing purpose, either from MU to AU (or) AU to MU.

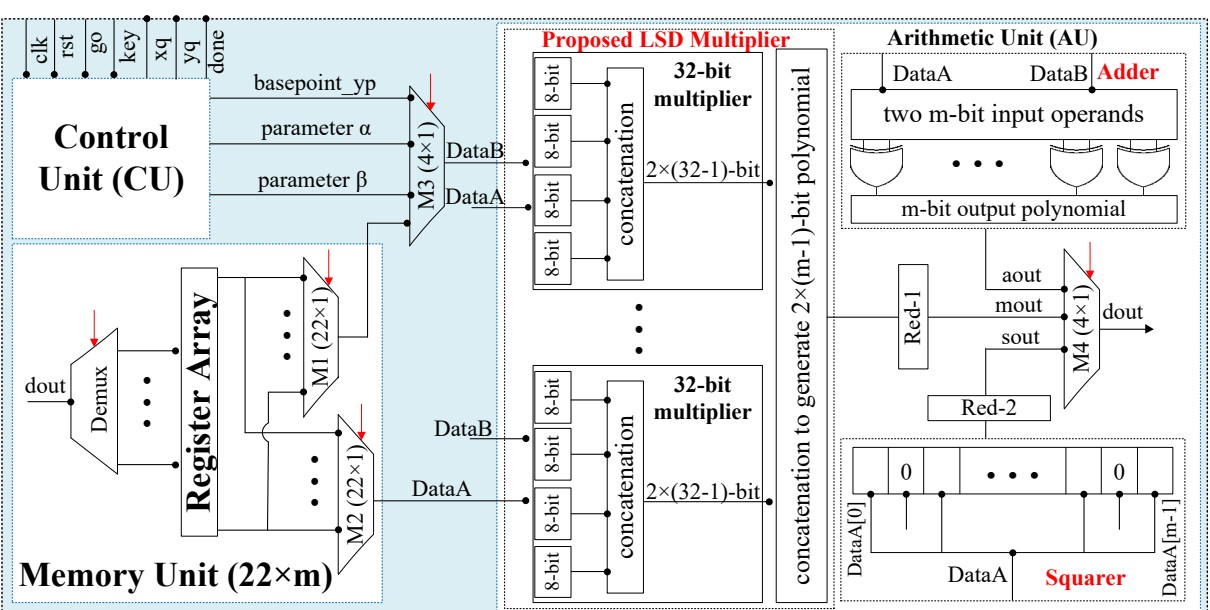

**Figure 1.** Proposed AREEBA architecture for PM on the BHC curve.

### 3.1. Memory Unit

An array of the register file is inferred as a memory unit, as given in Figure 1. It contains a total of 22 addresses with a data size of *m*-bits. The variable *m* determines the size of field (or) the targeted key length (163 or 233). The objective of this unit is to, (1) maintain the initial BHC curve parameters, i.e., $x_1, x_2, y_1, y_2, z_1, z_2$, (2) contain the intermediate results while implementing Algorithm 1 and (3) store the *x* and *y* coordinates of a final BHC curve point, i.e., $x_q, y_q$. In order to read two operands from MU in one clock cycle, two multiplexers ($M_1$ and $M_2$) are required. A single demultiplexer (*Demux*) is also utilized to update the computed result on the specified MU address.

### 3.2. Arithmetic Unit

As shown in Figure 1, the required building blocks for the PM computation on BHC curve are an adder, a squarer, a multiplier and two reduction units. The corresponding details to implement these building blocks are given in the following text:

- **Adder:** As compared to $GF(\mathbb{P})$, the selected $GF(2^m)$ field provides a carry save addition. However, the adder unit in this work is implemented by performing a bitwise Exclusive-OR operation, as shown in Figure 1. For two $m$-bit input operands, $m$ number of Exclusive-OR gates are required to produce an $m$-bit resultant polynomial after addition.

- **Squarer:** As provided in Section 1.2, the proposed architecture is developed to implement PM computation using a polynomial representations. The polynomial representation offers squaring computation by inserting an extra bit (i.e., 0) after each input data bit, as given in Figure 1. The squarer unit takes an $m$-bit operand as input and results a polynomial of length $(2 \times m - 1)$-bits after squaring.

- **Multiplier:** A general overview of the proposed 32-bit segmented LSD multiplier is shown in Figure 1. Similarly, the detailed architecture of the proposed multiplier is presented in Figure 2. It consists of four units, i.e., polynomial segmentation and digit creation, digit multiplication, internal concatenation and the final concatenation. The description of these units is given as follows:

  - **Polynomial segmentation and digits creation:** In the proposed LSD multiplier architecture, the first operand (Poly$_A$) is considered as $m$-bit. Similarly, the second operand (Poly$_B$) of 32-bit is created. The total number of digits for an $m$-bit polynomial is calculated using $n = \frac{m}{d}$. The $n$ determines the total digits (i.e., digit-$B_1$ to digit-$B_8$). On the other hand, the $m$ and $d$ show the operand length and digit size, respectively. As shown in Figure 2, each 32-bit digit is partitioned into four 8-bit segments. These segments are $B_{1,1}$, $B_{1,2}$, $B_{1,3}$ and $B_{1,4}$. In the subscript of $B_{1,4}$, the first integer (i.e., 1) determines the number of digits. Similarly, the second integer (i.e., 4) shows the number of segments. The same concept is employed to identify the created digits and the partitioned segments.

  - **Polynomial multiplication:** As shown in Figure 2, an $m$-bit Poly$_A$ and the created 8-bit segments of the corresponding digit are input to a schoolbook polynomial multiplier for multiplication. Therefore, it results four $(8 + m - 1)$-bit polynomials as output (the length of the polynomials is not shown in Figure 2). We are thankful to the authors of [28], for sharing their open source repository of various polynomial multipliers, available at https://github.com/Centre-for-Hardware-Security/TTech-LIB, accessed on 8 May 2021. According to our requirements, we generated the Verilog (HDL) design of an $8 \times m$-bit schoolbook polynomial multiplier by using the open source generator, provided by [28].

  - **Internal concatenation:** The multiplication, after each 8-bit segment to Poly$_A$, results in a polynomial of length $8 + m - 1$-bit. For each 32-bit digit multiplication, a concatenation over segmented polynomials of length $8 + m - 1$-bit is performed to produce a resultant polynomial of length $32 + m - 1$-bit.

  - **Final concatenation:** Once the multiplication over each 32-bit digit is completed, the final polynomial of length $2 \times m - 1$-bit is generated using a concatenation of $(32 + m - 1)$-bit polynomials.

- **Reduction:** After each polynomial multiplication and squaring, the resultant polynomials are $2 \times m - 1$-bit. Therefore, a reduction operation is required to reduce $2 \times m - 1$-bit polynomials to $m$-bit. In this work, the reduction units (Reduction-1, connected serially after the LSD multiplier and Reduction-2, connected serially after the squarer unit) are implemented by considering the NIST recommended field reduction algorithms over $GF(2^{163})$ and $GF(2^{233})$, respectively. For reduction algorithms, we refer interested readers to Algorithm 2.41 and Algorithm 2.42 from [29].

- **Inversion:** As shown in Algorithm 1, the inversion computation is required to perform reconversions from the projective to affine coordinated systems. To compute FF inversion, an Itoh–Tsujii algorithm [30] is used in this work. It requires only the multiplication and squaring operations. Consequently, the inversion is implemented by using hardware resources of an LSD multiplier and a squarer unit.

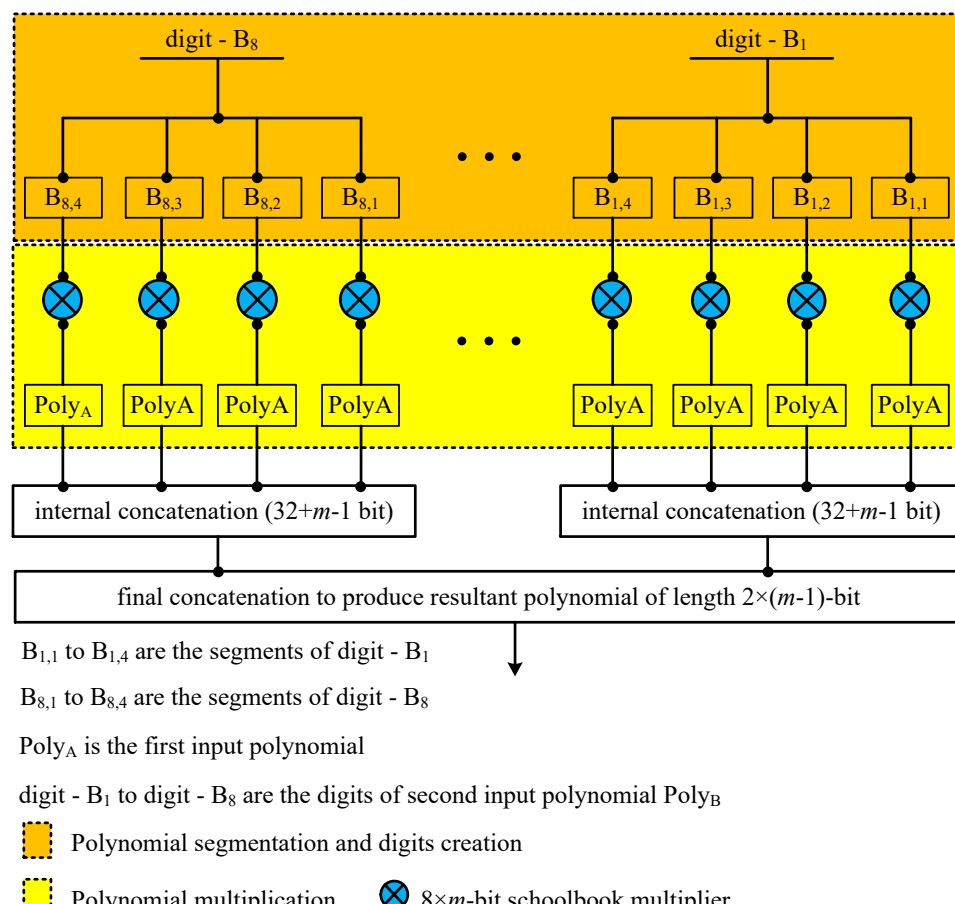

$B_{1,1}$ to $B_{1,4}$ are the segments of digit - $B_1$

$B_{8,1}$ to $B_{8,4}$ are the segments of digit - $B_8$

$Poly_A$ is the first input polynomial

digit - $B_1$ to digit - $B_8$ are the digits of second input polynomial $Poly_B$

Polynomial segmentation and digits creation

Polynomial multiplication ⊗ 8×*m*-bit schoolbook multiplier

**Figure 2.** Detailed architecture of the proposed segmented-LSD multiplier over $GF(2^{233})$.

### 3.3. Control Unit

The control functionalities are performed by employing an efficient finite state machine (FSM)-based dedicated controller. The dedicated FSM controller is termed as CU in Figure 1. The CU generates signals for routing multiplexers as well as the read and write addresses for MU. The used control signals are shown with red color lines in Figure 1. The corresponding FSM states (St) for the generation of these signals are also given.

In order to implement Double and Add algorithms (Algorithm 1), the FSM incorporates a total of 88 states (St: 0–St: 87), as shown in Figure 3. The St: 1–St: 3 are required to generate control signals for the initialization part. The corresponding PD and PA states are St: 4–St: 34 and St: 35–St: 65, respectively. The last states (i.e., St: 34 and St: 65) of each PD and PA operation are responsible for monitoring the inspected key bit (*Key*). Additionally, it also counts the number of points on the specified BHC curve using a signal *count*. Once the value for *Key* = 0 and the value for *count* ≠ 0, the processor returns to St: 4 from St: 34 and St: 65. Whenever, the value for *count* = 0, the processor returns to St: 66. Finally, St: 66–St: 86 generate control signals for the reconversions process, including an FF inversion operation. The additional two states (St: 85 and St: 86) are used to implement the remaining FF operations of Algorithm 1.

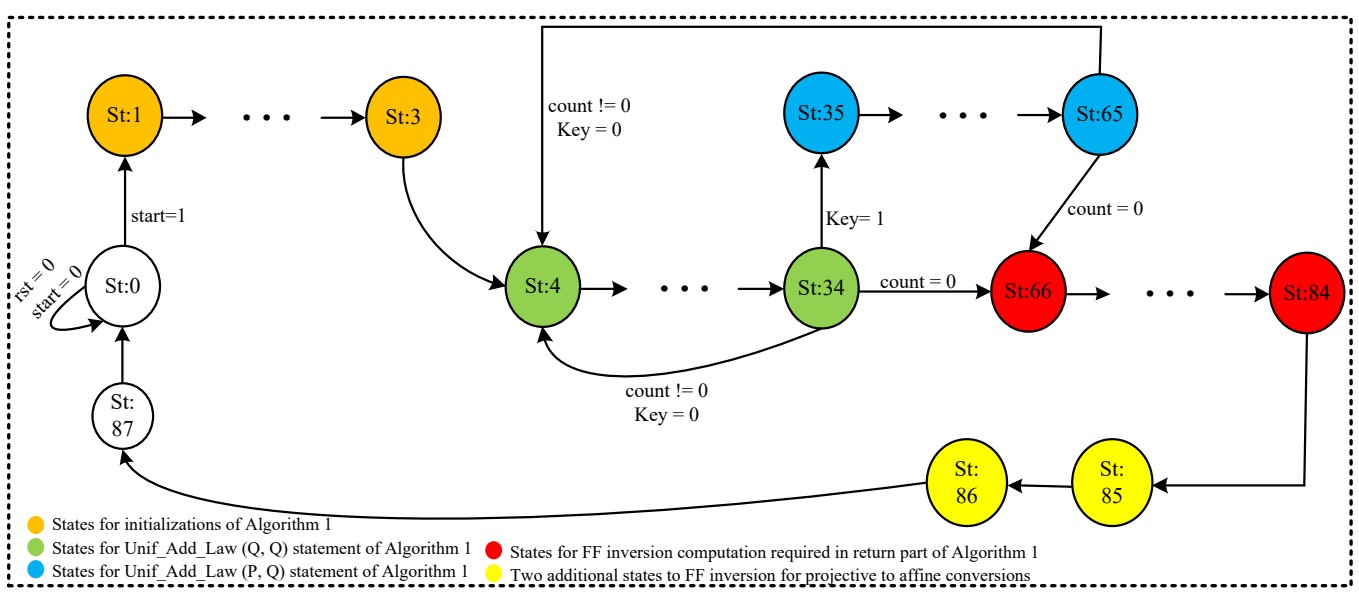

**Figure 3.** FSM of the proposed architecture.

## 4. Implementation Results and Comparisons

This section first provides parameters for the implementation of selected BHC model in Section 4.1). Subsequently, the synthesis results for various FPGA devices are given in Section 4.2. Finally, the comparison with existing state of the art solutions is provided in Section 4.3.

### 4.1. Implementation Parameters

The required parameters for the PM implementation of BHC over $GF(2^{163})$ and $GF(2^{233})$ are shown in Table 3. The parameters $Basepoint_{xp}$, $Basepoint_{yp}$, $Constant_a$ and $Constant_b$ are selected from [27]. The values for $\alpha$ and $\beta$ parameters are calculated using $\frac{(a+b)}{b}$ and $\frac{(a+b)}{a}$, respectively. For an $m$-bit key length, $\frac{m}{2}$ unified PD and PA computations are performed. Moreover, the scalar multiplier ($Key$) is selected randomly alternating between 0 s and 1 s.

**Table 3.** Implementation parameters over $GF(2^{163})$ and $GF(2^{233})$ [27].

| Key Length ($m$) | Selected Parameters and the Corresponding Values |
|---|---|
| $GF(2^{163})$ | $Basepoint_{xp} = 3f0eba16286a2d57ea0991168d4994637e8343e36$<br>$Basepoint_{yp} = 0d51fbc6c71a0094fa2cdd545b11c5c0c797324f1$<br>$Constant_a = 00000000000000000000000000000000000000001$<br>$Constant_b = 20a601907b8c953ca1481eb10512f78744a3205fd$<br>$\alpha = 0fbdc5250853a4db63fe6163034e298c7805300bc$<br>$\beta = 000000000000000000000000000000000004a3205fc$<br>$Key = 5555555555555555555555555555555555555555$ |
| $GF(2^{233})$ | $Basepoint_{xp} = 0fac9dfcbac8313bb2139f1bb755fef65bc391f8b36f8f8eb7371fd558b$<br>$Basepoint_{yp} = 1006a08a41903350678e58528bebf8a0beff867a7ca36716f7e01f81052$<br>$Constant_a = 00000000000000000000000000000000000000000000000000000000001$<br>$Constant_b = 066647ede6c332c7f8c0923bb58213b333b20e9ce4281fe115f7d8f90ad$<br>$\alpha = 00000000000000000000fbdc5250853a4db63fe6163034e298c7805300bc$<br>$\beta = 00000000000000000000000000000000000000000000000000004a3205fc$<br>$Key = 55555555555555555555555555555555555555555555555555555555555$ |

### 4.2. Implementation Results

We modeled two Verilog (HDL) designs over $GF(2^{163})$ and $GF(2^{233})$ using the Xilinx ISE design tool. To perform functional verification, the behavioral simulation models are verified with their corresponding C-based functional implementations. The implementation results for various Xilinx FPGA devices are given in Table 4. For Virtex-4, Virtex-5, Virtex-6

and Virtex-7 FPGA boards, the selected devices for logic synthesis are xc4vfx140-11ff1517, xc5vfx130t-3ff1738, xc6vlx550t-2ff1760 and xc7vx690t-3ffg1930, respectively. Column one in Table 4 presents the key length (m) while the implementation platform is given in the second column. The remaining columns (three to six) provide FPGA slices, operational clock frequency (in MHz), number of clock cycles and latency (in μs), respectively. The values for reported FPGA slices and clock frequency are determined with the Xilinx ISE tool. Similarly, the number of clock cycles and latency values are calculated using Equations (3) and (4), respectively.

$$\text{Clock cycles} = \frac{\text{time required for behavioral simulations}}{\text{time period}} \tag{3}$$

$$\text{Latency (in } \mu\text{s)} = \frac{\text{Clock cycles}}{\text{Frequency (in MHz)}} \tag{4}$$

**Table 4.** Implementation results for $GF(2^{163})$ and $GF(2^{233})$ on Xilinx Virtex-4, 5, 6 and 7 FPGA devices.

| Key Length (m) | Platform | Slices | Frequency (in MHz) | Clock Cycles | Latency (in μs) |
|---|---|---|---|---|---|
| 163 | Virtex-4 | 5302 | 152 | 10,393 | 68 |
| | Virtex-5 | 2412 | 194 | 10,393 | 53 |
| | Virtex-6 | 2982 | 200 | 10,393 | 51 |
| | Virtex-7 | 3508 | 242 | 10,393 | 42 |
| 233 | Virtex-4 | 11,557 | 103 | 11,137 | 108 |
| | Virtex-5 | 10,065 | 157 | 11,137 | 70 |
| | Virtex-6 | 4370 | 164 | 11,137 | 67 |
| | Virtex-7 | 4261 | 204 | 11,137 | 54 |

As shown in Table 4, the proposed architecture over $GF(2^{163})$ and $GF(2^{233})$ requires 10,393 and 11,137 clock cycles, respectively. For each implementation platform over $GF(2^{163})$ and $GF(2^{233})$, the achieved results in terms of FPGA slices, clock frequency and time to perform one PM operation are given in the following:

**Results on Virtex-4 and Virtex-5.** As shown in column three of Table 4, the proposed architecture over $GF(2^{163})$ and $GF(2^{233})$ utilizes 5302 and 11,557 slices on Virtex-4. Moreover, the achieved clock frequency is 152 and 103 MHz, respectively. Apart from hardware slices and clock frequency, the time required to perform one PM computation is 68 μs and 108 μs. On Virtex-5, the proposed architecture over $GF(2^{163})$ and $GF(2^{233})$ utilizes 2412 and 10,065 FPGA slices which are comparatively 1.14 and 2.19 times lower than our Virtex-4 slices. The achieved clock frequency increases as compared to our Virtex-4 implementations. The increase in clock frequency (from 152 to 194 over $GF(2^{163})$ and from 103 to 157 over $GF(2^{233})$) ultimately reduces the latency, as shown in Table 4.

**Results for Virtex-6 and Virtex-7.** On Virtex-6, the utilized FPGA slices over $GF(2^{163})$ and $GF(2^{233})$ are 2982 and 4370, respectively. For the same key lengths (163 and 233), the achieved clock frequency is 200 MHz and 164 MHz. When moving to Virtex-6 from Virtex-5 FPGA, there is a small decrease in the computation of PM time (53 μs to 51 μs over $GF(2^{163})$ and 70 μs to 67 μs over $GF(2^{233})$). On newer Virtex-7 FPGA, the proposed architecture achieves higher clock frequency as compared to our implementations on Virtex-4, Virtex-5 and Virtex-6 devices. For a higher key length ($m = 233$), the proposed architecture consumes lower hardware slices as compared to the corresponding Virtex-4, Virtex-5 and Virtex-6 FPGA implementations. The required computation time for one PM is 42 μs and 54 μs over $GF(2^{163})$ and $GF(2^{233})$, respectively.

To summarize, the proposed architecture consumes lower slices on Virtex-5 and Virtex-7 devices. The newer technologies (Virtex-6 and Virtex-7) provide a relatively higher clock frequency as compared to older Virtex-5 and Virtex-4 FPGAs. As shown in the last column of Table 4, the latency increases with an increase in the key length. Moreover, as the target platform changes from Virtex-4 to Virtex-7, the latency of the architecture is decreased.

### 4.3. Comparison with State of the Art Architectures

In order to provide a realistic and reasonable comparison with state of the art, we synthesized our Verilog (HDL) models for similar FPGA devices, as shown in Table 5. The comparison with a variety of existing low area architectures of BHC model is challenging as there are fewer hardware-based published works [18–22]. Therefore, we also provided a comparison with the most recent area optimized implementations of the unified BEC model [8,23–25]. It is important to mention that we have placed the symbol '−' in Table 4 where the relevant information is not given.

**Table 5.** Comparison with state of the art architectures over *Unif_Add_Law* computations for PM.

| Ref #. | Key Length (m) | Platform | Slices | Frequency (in MHz) | Clock Cycles | Latency (in μs) |
|--------|---------------|----------|--------|--------------------|--------------|-----------------|
| **Architectures for the unified BHC model of ECC** | | | | | | |
| [18] | $GF(2^{233})$ | Virtex-4 | 20,437 | 81 | 5913 | 73 |
| [21] | $GF(2^{233})$ | Virtex-4 | 9763 | 329 | 13,057 | 39 |
| [21] | $GF(2^{233})$ | Virtex-5 | 6703 | 397 | 13,057 | 32 |
| [19] | $GF(2^{233})$ | Virtex-6 | 7150 | 172 | 7370 | 43 |
| [20] | $GF(2^{233})$ | Virtex-6 | 7681 | 296 | 11,838 | 39 |
| [19] | $GF(2^{233})$ | Virtex-7 | 6032 | 183 | 7370 | 40 |
| [20] | $GF(2^{233})$ | Virtex-7 | 6342 | 369 | 11,838 | 32 |
| [21] | $GF(2^{233})$ | Virtex-7 | 7017 | 434 | 13,057 | 30 |
| [22] | $GF(2^{233})$ | Virtex-7 | 6083 | 341 | 12,553 | 36 |
| **Architectures for the unified BEC model of ECC** | | | | | | |
| [23] | $GF(2^{233})$ | Virtex-4 | 21,816 | 48 | − | − |
| [24] | $GF(2^{233})$ | Virtex-5 | 5919 | − | − | 26.24 |
| [25] | $GF(2^{233})$ | Virtex-5 | 15,804 | 308 | − | − |
| [8] | $GF(\mathbb{P})$ | Virtex-6 | 6600 | 93 | − | 2130 |
| **AREEBA** | $GF(2^{233})$ | Virtex-4 | 11,557 | 103 | 11,137 | 108 |
| | $GF(2^{233})$ | Virtex-5 | 10,065 | 157 | 11,137 | 70 |
| | $GF(2^{233})$ | Virtex-6 | 4370 | 164 | 11,137 | 67 |
| | $GF(2^{233})$ | Virtex-7 | 4261 | 204 | 11,137 | 54 |

**Comparison with BHC and BEC architectures on Virtex-4:** The BHC and BEC architectures on Virtex-4 FPGA are reported in [18,21,23], respectively. As shown in Table 5, our design consumes 1.76 times fewer hardware slices over Virtex-4 as compared to [18]. This is due to the use of multiple FF operators (multiplier and adder) in the datapath. On the other hand, the proposed architecture utilizes only one segmented-LSD multiplier, adder and squarer in the datapath. Additionally, the use of a hybrid Karatsuba multiplier (by merging general and simple multipliers) increases hardware resources. Moreover, the achieved operational clock frequency in our design is 103 MHz which is comparatively 1.27 times higher. However, to perform one PM computation, it requires more clock cycles and needs higher computational time (in terms of latency). The architecture of [21] utilizes 1.18 times lower hardware slices as compared to this work. Nevertheless, the proposed architecture requires 1.17 times lower clock cycles. It implies that there is always a trade-off between the achieved performance and the consumed area. The BEC architecture, presented in [23], utilizes 1.88 times more hardware resources compared to our architecture. This is due to use of the hybrid Karatsuba multiplier in the datapath. In addition to the optimized hardware resources, the presented architecture also provides 1.68 times higher clock frequency.

**Comparison with BHC and BEC architectures on Virtex-5:** The BHC and BEC architectures on Virtex-5 FPGA are reported in [21,24,25], respectively. The architecture in [21] utilizes 1.50 times fewer slices as compared to this work. Similarly, the BEC architecture in [24] consumes 1.70 times fewer slices. The comparison in terms of clock cycles and frequency is not possible as the values for these design parameters are not given. The BEC architecture in [25] utilizes 1.57 times more hardware resources as compared to our design. However, our design provides 1.96 times lower clock frequency.

**Comparison with BHC and BEC architectures on Virtex-6:** The BHC and BEC architectures on Virtex-6 FPGA are reported in [8,19,20], respectively. In ref. [19], a hybrid Karatsuba multiplier is employed. The use of a segmented-LSD multiplier in the proposed architecture results in

1.63 times fewer slices. Furthermore, the architecture of [19] requires 1.51 times fewer clock cycles and requires lower computational time. A digit-parallel least significant digit multiplier, with a digit size of 32-bit, is incorporated in [20]. The use of a digit parallel multiplier results in 1.75 times more hardware resources. For the twisted BEC, the prime $GF(\mathbb{P})$ with $\mathbb{P} = 233$ is utilized in [8]. The proposed architecture utilizes 1.51 times fewer slices as compared to the most recent BEC architecture of [8]. Moreover, for same the key lengths (i.e., 233), the proposed architecture over $GF(2^m)$ is 31.79 (ratio of 2130 over 67) times faster.

**Comparison with BHC architectures on Virtex-7:** The BHC architectures on Virtex-7 FPGA are reported in [19–21]. The use of a segmented-LSD multiplier in this article results in 1.41 times lower slices as compared to [19], where an hybrid Karatsuba multiplier is employed. Furthermore, an increase of 1.11 times in operational clock frequency is also obtained. Similarly, the use of a digit parallel multiplier in [20] results in 1.48 times more hardware resources. Due to four-stage pipelining, the architecture of [21] achieves a higher clock frequency. Nevertheless, the reported clock cycles are 1.17 times higher than this work. This is due to the inherent data dependency in the $Unif\_Add\_Law$ of BHC model (see Table 2). Using the same FF multiplier of [20], the dedicated architecture in [22] consumes 1.42 times more slices. Moreover, the architectures of [20,22] achieve a higher operational clock frequency. On the other hand, the proposed solution in this article requires fewer clock cycles.

## 5. Conclusions

This article has provided an area optimized hardware accelerator for the PM computation of BHC model over $GF(2^{163})$ and $GF(2^{233})$. The area optimization is achieved by employing a segmented-LSD multiplier in the datapath of the proposed hardware accelerator. To provide a realistic and reasonable comparison with state of the art, the proposed architecture is synthesized for various FPGA devices. For Virtex-4, Virtex-5, Virtex-6, and Virtex-7 devices, the utilized FPGA slices over $GF(2^{163})$ are 5302, 2412, 2982 and 3508, respectively. On similar FPGA devices over $GF(2^{233})$, the achieved values are 11,557, 10,065, 4370 and 4261, respectively. It is important to note that the use of the segmented-LSD multiplier results in fewer hardware resources on newer FPGA devices (Virtex-6 and Virtex-7). On the other hand, the proposed low area accelerator architecture requires a slightly higher number of clock cycles for the PM computation. In other words, there is always a trade-off between achieved performance and consumed area. Therefore, the reported low area values prove the acceptability of this work in area-constrained applications.

**Author Contributions:** Conceptualization, M.I. and S.S.J. and M.R.; data extraction, A.S. and S.S.J.; results compilation, A.S. and S.S.A., M.H.S. and M.R.; validation, M.I. and M.R.; writing—original draft preparation, A.S. and M.I.; critical review, M.I. and M.R. and S.S.J.; draft optimization, A.S.; supervision, M.I. and M.R.; funding acquisition, S.S.J. All authors have read and agreed to the published version of the manuscript.

**Funding:** The author Sajjad Shaukat Jamal extends his gratitude to the Deanship of Scientific Research at King Khalid University for Funding this work through research group program under grant number R.G.P. 1/72/42.

**Acknowledgments:** We are thankful to the support of King Abdul-Aziz City for Science and Technology (KACST) and Science and Technology Unit (STU), MAKKAH, Saudi Arabia.

**Conflicts of Interest:** The authors declare no conflict of interest.

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
