# Peer review of "AREEBA: An Area Efficient Binary Huff-Curve Architecture"

_electronics, doi:10.3390/electronics10121490_

Round 1

Reviewer 1 Report

The authors propose a FPGA implementation of operations on elliptic curves (point addition and doubling). The authors select one of the possible models that give support to the framework, pointing out that the selection is due to the higher security this model offers with respect to other (more efficient) ones.

The paper contains interesting results. Nevertheless, I have got some concerns that I enumerate below. The manuscript is understandable but its English really needs polishing by a native English professor (please, find below a — non exhaustive — list of the reasons that alllow me to conclude this).

Main concerns:

The authors claim that BHC model is more secure “(it) has higher computational complexity as compared to BEC and HC models **which ultimately results in higher security**”. Despite the authors provide a citation that (supposedly) endorse this claim, I really do not see the causality of the statement. The authors should **clearly and undoubtedly** substantiate this. 

It seems to me that the authors identify security with hardware implementation. In my honest opinion, there are as many as perils in pure hardware implementation than in software ones. Furthermore, the lack of hardware implementation for a given model should not suffice for removing other models from the experimentation. In order to prove validity of the results, the authors should provide not only good experimental results with respect to previous results on the very same framework, but also with respect to other different approaches (at least a sound justification must be provided). The argument provided in the beginning of Section 4.3 does not suffice (furthermore, when the cited bibliography refers almost exclusively to previous results of the same research team).

In the conclusions, the authors claim that their use of a segmented LSD multiplier implies the need of less hardware needs, but also that, “as expected, the proposed architecture requires higher (sic) clock cycles for PM computation”. In my opinion this is a severe drawback since the architecture is much slower than others. The lower hardware requirements do not justify the lack of performance.  At least the authors should provide better and sound arguments.

Other concerns:

Section 1.2 Contributions, contains nothing to do with real contributions of the paper: first one list authors’ results in other papers; second and third provide an argument to use BHC model (which is not a contribution); and, fourth explains some detail of the proposed architecture which would find better located in the description of the proposed model.

The authors state that AREEBA stands for “Area Efficient Binary Huff-curve Architecture”. I really do not see the acronym. It is really not important, the authors can name their architecture as they want, but if they provide an explanation it should fit with the acronym.

Lines 125-130. The authors explain the experimentation carried out and some results. References to experimental results in this section should be limited to a brief comparative description with respect to other models (“it performs better but”, it outperforms previous results”, …)

Algorithm 1 describes a typical exponentiation process by successive doubling with the only inclusion of some formulae related to BHC model. IMO it is not necessary and it would suffice a reference to the modification, but I leave this to the authors consideration.

Line 166: I guess that the equation should be Q= (P + P + … + P) = kP instead of  Q= k x (P + P + … + P) = kP. Maybe it is not the notation I am used to. Please clarify.

Results in Sections 4.2 and 4.3 should be better organized by platform, since it seems that the main comparison of the results considers that feature.

(Non-exhaustive) list of English flaws:

Lines 52-54: please rephrase. It is usually more readable a sentence similar to “The main objective of this work is…”. Bold type is unnecessary, please get rid of it.

Lines 56-58: please rephrase. I date to suggest something in the vein of “In [15-19], some new architectures are proposed to provide resistance against SAP to BEC and HC models” 

Line 149: The authors [16] *have* identified that the unified PA… get rid of the *have* or rephrase.

Line 162-163: (non exhaustive) references to “SPA attacks” is redundant since I see that SPA stands for “Simpe Power Attacks”.

Line 183: (I guess something like the following suits best) The objective*s* of this unit are to: (i) … ; (ii) … ; (iii) …

Please, carry out a thoroughly English review of the paper (special attention to Sections 4.2 and 4.3).

Reviewer 2 Report

The article addresses the implementation of a low chip area ECC processor. My comments are as follows:

1. It seems that the article contains 5 self-citations. If possible, these could be reduced at 2-3. 

2. Some articles have DOI content while others don't.

3. References [17] and [23] need more details, such as the source or title. The authors are invited to check the references.

4. There is an interesting attention granted to simple power analysis (SPA), yet we also have DPA and CPA, the last one being more effective against cryptoprocessors. The authors are invited to extend their discussion with DPA and CPA (case studies, if they work in the ECC case, etc). SPA is too simple to worth a dedicated attention. 

5. Also related to 3, it is not clear whether the proposed solution is SPA resistant.

6. I think Algorithm 1 deserves a distinct table, e.g. Table 3.

7. The font size used in Figure 2 is too large compared with Figure 1, they should have similar font size. 

8. "PM" is not defined in Abstract (line 6).

9. It is hard to believe that "Unified PD and PA" can be used as a distinct keyword.

10. "Crypto hardware" cannot be keyword, maybe "cryptoprocessor" (!?). In addition, why not inserting FPGA as a keyword since the implementation addresses FPGA implementations.

11. Weierstrass and Karatsuba are always used with capital letters, which is not the case for lines 42, 66, 67, 298, 299 etc. The authors are invited to check the manuscript.

12. "an efficient scheduling of PA and PD instructions are provided" is not clear (lines 84-85).

13. "area reduced architecture" is not so spread, we could use low chip area implementations/architecture. The same comment is applicable to "area critical crypto applications" and "for prevention from SPA" /line 182/ ("hardening" sounds better).

14. Mixing text and mathematical symbols is not welcome, this is the case of line 143 (I suppose the curve parameters belong to ...).

15. "The value of" is not welcome as well (line 182 and many others)

16. "the size of digit" is not welcome, we could use digit size (even though it is still strange).

17. "can be accessed from [23]" (line 232) sounds strange as well.

18.  I think that latency decreases, instead of "is decreased" (line 285). Also "the latency increases with the increase in m" does not sound so well.

19. The phrase spread on lines 285-286 should be rephrased.

20. Normally we have a design occupying 1.76 times fewer FPGA slices (line 294).

21. I think clock cycles and latency are given by (3) and (4), not calculated.

22. The FSM given in Figure 3 is not clear, especially the states 4 -> 34, 66 -> 84 and 35 -> 65, it is not obvious what processes/instructions are executed in-between states.

23. Comparing our own performances with a singular reference, phrase by phrase, is not welcome, in fact I believe it is not politely at all. Normally, we should resume our performances and insert a table for comparison. There is no need to emphasize our unprecedented performances. This is the case of section 4.3 where [15] is mentioned 6 times, [17] 5 times, [18] and [16] 4 times.

Reviewer 3 Report

Topics such as those covered in the manuscript are having a significant impact on the international scientific community and achieving excellent results. This, of course, provides an extra gear to the paper, which is in itself very interesting, clear in the exposition and well organized. However, there are numerous inaccuracies, not just textual, and some parts that could be improved or implemented.

The following, for example, is a mixed, NOT exaustive, list of observations. I suggest to the authors to check carefully the whole paper to look for other imperfections.

(1) Page 2, lines 33 and 34: use capital letters when needed. For example, Weierstrass, Edward, etc.

(2) Page 2, line 42: as in (1).

(3) Page 2, line 67: "karatsuba" -> "Karatsuba". Check the whole manuscript (e.g., page 10 line 298).

(4) Page 3, line 79: "MHz" is a measure unit, hence it is usually written in roman characters. The same for "\mu s" in the line 95. Check the whole manuscript, also the tables.

(5) Page 3, line 92: "Weirstrass" -> "Weierstrass".

(6) Page 4, line 141: is "The binary Huff curve" better? "by satisfying" -> "satisfying" or "which satisfy the following equation:" (remove "1").

(7) Page 4, line 143: "equation 1" -> "Equation (1)" or "Eq. (1)". "\in" -> "belong to".

(8) Page 5, line 154: "are the projective points" -> "are the projective coordinates of the points".

(9) Page 5, line 155: remove the comma after \alpha. Moreover, something like "are constants defined as" sounds better than "the curve constant parameters and computed".

(10) Page 5, line 163: "considers".

(11) Page 5, line 168: replace "by repeating the $k$ times additions, as given in equation 2:" with something like "by iterating $k$ times the sum of a point $P$ as below:".

(12) Page 5, Eq. (2): write it in a better way, for example,

Q=k\cdot P\ :=\ \underbrace{P+P+\ldots+P}_{k\text{-times}}

(13) Page 5, line 167: "equation 2" -> "Equation (2)" or "Eq. (2)".

(14) Page 5: in the table of Algorithm 1 (or after it), explain better the meaning of "k=(k_{n-1},\ldots,k_1,k_0)". In the same line, P=(x,y) should belong to (GF(2^m))^2.

(15) Page 6, line 192: the symbol "GF" (Galois field) is usually/preferably written in roman (see the typographical conventions in mathematical formulae, also on Wikipedia). Instead "p" in "GF(p)" has the appropriate math/slanted font.

(16) Page 7, line 199: add a couple of round brackets; try, for example, $(2\times m-1)$-bits

(17) Page 7, line 213: add a full stop after "digits".

(18) Page 7, line 218: add a couple of round brackets as in (16). In other cases, consider if it is better just to write, for example, "length $8+m-1$ bits" (see lines 221-223).

(19) Page 9, Table 3: the exadecimal characters a-f are usually written in roman because they are not mathematical variables.

(20) Page 10, Eqs. (3) and (4): here you should use text characters (and not mathematical fonts).

_______________________

Sometimes the information provided could be improved for the benefit of the reader. For example, there are no general references, specialized monographs and reviews in the bibliography. Not even articles in contiguous fields that may represent important future prospects. You could choose some recent reviews or articles with an extensive guided bibliography and refer to them: for example {doi: 10.1007/s00009-020-01531-5} discusses many theoretical sources and references on innovative supercomputing techniques applied to curves and more, which seem to be interesting also from a cryptographic-quantum perspective. Another example: about Hessian, Huff and Edward curves you can say much more, e.g., generalizations {doi: 10.1007/978-3-319-38898-4_2}, theoretical foundations {doi: 10.1090/s0273-0979-07-01153-6}, {doi: 10.1007/3-540-44709-1_11}, isogenies {doi: 10.1515/jmc-2020-0037}, etc. Continue in this way, looking, in particular, for new ideas and researches such as supercomputing in cryptography and similar things.

In conclusion, the paper needs minor/moderate revisions, after which I might recommend it for publication on Electronics.

Round 2

Reviewer 1 Report

I thank the authors for the extensive revision.